# Variational Adaptive Graph Transformer for Multivariate Time Series Modeling

## Abstract

Multivariate time series (MTS) are widely collected by large-scale complex systems, such as internet services, IT infrastructures, and wearable devices. The modeling of MTS has long been an important but challenging task. To capture complex long-range dynamics, Transformers have been utilized in MTS modeling and achieved attractive performance. However, Transformers in general do not well capture the diverse relationships between different channels within MTS and have difficulty in modeling MTS with complex distributions due to the lack of stochasticity. In this paper, we first incorporate relational modeling into Transformer to develop an adaptive **G**raph **Trans**former (G-Trans) module for MTS. Then, we further consider stochastity by introducing a powerful embedding guided probabilistic generative module for G-Trans to construct **V**ariational adaptive **G**raph **Trans**former (VG-Trans), which is a well-defined variational generative dynamic model. VG-Trans is utilized to learn expressive representations of MTS, being an plug-and-play framework that can be applied to forecasting and anomaly detection tasks of MTS. For efficient inference, we develop an autoencoding variational inference scheme with a combined prediction and reconstruction loss. Extensive experiments on diverse datasets show the efficient of VG-Trans on MTS modeling and improving the existing methods on a variety of MTS modeling tasks.

## 1 Introduction

Multivariate time series (MTS) is an important type of data that arises from a wide variety of domains, including internet services (Dai et al., 2021; 2022), industrial devices (Finn et al., 2016; Oh et al., 2015), health care (Choi et al., 2016b;a), and finance (Maeda et al., 2019; Gu et al., 2020), to name a few. However, the modeling of MTS has always been a challenging problem as there exist not only complex temporal dependencies, as shown in the red box in Fig. 1, but also diverse cross-channel dependencies, as shown in the blue box in Fig. 1. Moreover, there exist inherently stochastic components, as shown in the green box in Fig. 1, even if one can fully capture both temporal and cross-channel dependencies. To address these challenges, many deep learning based methods have been proposed for various MTS tasks, such as forecasting, anomaly detection, and classification.

To model the temporal-dependencies of MTS, many dynamic methods based on recurrent neural networks (RNNs) have been developed (Malhotra et al., 2016; Zhang et al., 2019; Bai et al., 2019b; Tang et al., 2020; Yao et al., 2018). Meanwhile, to take the stochasticity into consideration, some probabilistic dynamic methods have also been developed (Dai et al., 2021; 2022; Chen et al., 2020; 2022; Salinas et al., 2020). With the development of Transformer (Vaswani et al., 2017) and due to its ability to capture long-range dependencies (Wen et al., 2022; Dosovitskiy et al., 2021; Dong et al., 2018; Chen et al., 2021), and interactions, which is especially attractive for time series modeling, there is a recent trend to construct Transformer based MTS modeling methods and have achieved promising results in learning expressive representations for down-stream tasks. For example, for forecasting, LogTrans (Li et al., 2019) incorporates causal convolutions into self-attention layer to consider local temporal dependencies of MTS. Informer (Zhou et al., 2021) develops a probsparse self-attention mechanism for long sequence forecasting. AST (Wu et al., 2020) further constructs a generative adversarial encoder-decoder framework for better predicting output distribution. In addition, there are also some other efficient Transformer-based forecasting methods, such as Autoformer (Xu et al., 2021), FEDformer (Zhou et al., 2022), and TFT (Lim et al., 2021).

Besides, for anomaly detection, Meng et al. (2019) illustrate the superiority of using Transformer for anomaly detection over other traditional RNN-based methods. Following it, some modified Transformer-based methods have also been proposed for anomaly detection, such as TransAnomaly (Zhang et al., 2021), ADTrans (Tuli et al., 2022), and Anomaly Transformer (Xu et al., 2022). To address non-deterministic temporal dependence within MTS, Tang & Matteson (2021) further incorporate Transformer structure into state-space models and develop ProTrans.

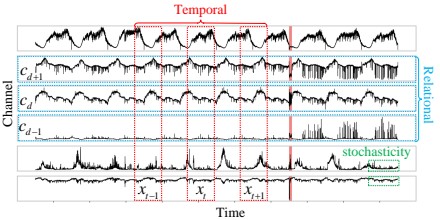

Figure 1: The temporal dependency, channel relationship and stochasticity within MTS.

Despite the attractive performance of existing Transformer-based models, their ultimate potentials have been limited by ignoring the cross-channel dependence of MTS.

To consider the relationships of different channels within MTS, MSCRED (Zhang et al., 2019) introduces a multi-scale convolutional recurrent encoder&decoder to learn spatial correlations and temporal characteristics in MTS and detects anomalies via the residual signature matrices. InterFusion (Li et al., 2021) incorporates recurrent and convolutional structures into a unified framework to capture both temporal and inter-metric information. Recently, Graph neural networks (GNNs) have gradually attracted more attentions in exploring the relationships. Thus, some GNN-based methods for MTS have been developed (Deng & Hooi, 2021; Zhao et al., 2020)for discovering expressive representations of MTS. Deep variational graph convolutional recurrent network (DVGCRN) incorporates relationship modeling into hierarchical generative process. Moreover, some graph based methods (Li et al., 2018; Bai et al., 2019a; Yu et al., 2018; Wu et al., 2019; Guo et al., 2019; Pan et al., 2019) have also been developed for MTS forecasting. Adaptive graph convolutional recurrent network (AGCRN) (Bai et al., 2020) further learns node-specific patterns for MTS forecasting without requiring a pre-defined graph. However, these methods are still all non-dynamic or RNN based models, limiting their power in capturing complex relationships across long-distance time steps

Moving beyond the constraints of previous work, we first propose an adaptive graph Transformer (G-Trans) module by incorporating a graph into the Transformer structure, which can model both temporal and cross-channel dependencies within MTS. Then, considering the stochasticity within MTS and enhancing the representative power of G-Trans, we further develop a **V**ariational adaptive **G**raph **Trans**former (VG-Trans), which is a well-defined probabilistic dynamic model obtained by combining G-Trans with a proposed **E**mbedding-guided **P**robabilistic generative **M**odule (EPM), as illustrated in Fig. 2 (b). We note that VG-Trans is able to get the robust representations of MTS, which enables it to be combined with the existed methods and applied to both anomaly detection and forecasting tasks. In addition, we introduce an autoencoding variational inference scheme for efficient inference and a joint optimization objective that combines forecasting and reconstruction loss to ensure the expressive time-series representation learning. The main contributions of our work are summarized as follows:

- For MTS modeling, we propose a G-Trans module, which incorporates channel-relationship learning into the Transformer structure.

- We develop VG-Trans, a VAE-structured probabilistic dynamic model with G-Trans as encoder and EPM as decoder, which can consider the non-deterministic within both temporal and cross-channel dependencies of MTS. VG-Trans can be combined with different methods and applied to different tasks of MTS.

- To achieve scalable training, we introduce an autoencoding inference scheme with a combined prediction and reconstruction loss for enhancing the representation power of MTS.

- Experiments on both anomaly detection and forecasting tasks illustrate the efficiency of our model on MTS modeling

## 2 METHOD

We first present the problem definition, and then introduce the probabilistic channel embedding for measuring the relationships between different channels and present G-Trans by incorporating cross-channel dependence into Transformer. Finally, we develop VG-Trans, a novel variational dynamic model. The notations used in this paper are summarized in Table 4 in Appendix.

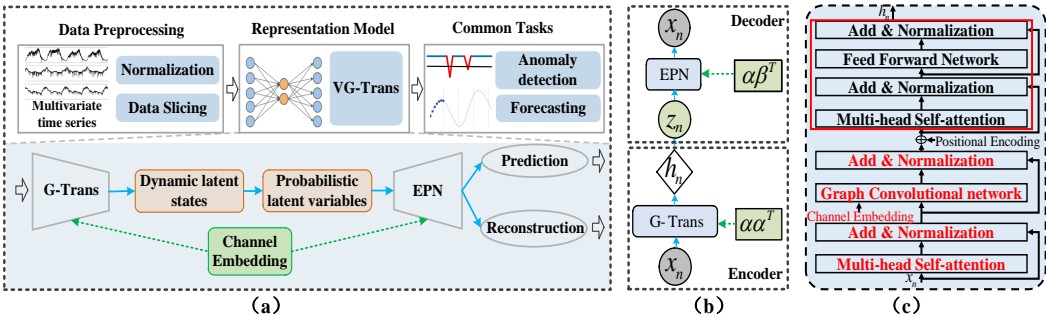

Figure 2: (a) The whole framework of VG-Trans for MTS modeling; (b) Graphical illustration of inference (encoder) and generative (decoder) models (Blocks in gray and green represent input and probabilistic latent variables, while the colourless blocks represents deterministic dynamic latent states.); (c) The detailed structure of adaptive graph Transformer module.

**Problem Definition:** Defining the $n$-th MTS as $\boldsymbol{x}_n = \{\boldsymbol{x}_{1,n}, \boldsymbol{x}_{2,n}, ..., \boldsymbol{x}_{T,n}\}$, where $n = 1, ...N$ and $N$ is the number of MTS. $T$ is the duration of $\boldsymbol{x}_n$ and the observation at time $t$, $\boldsymbol{x}_{t,n} \in \mathbb{R}^V$, is a $V$ dimensional vector where $V$ denotes the number of channels, thus $\boldsymbol{x}_n \in \mathbb{R}^{T \times V}$. The modeling of MTS is to learn the robust representations of the input with a powerful method, which can be served as a plug and play framework and applied to different tasks.

## 2.1 PROBABILISTIC CHANNEL EMBEDDING

To reflect the characteristics of different channels in MTS and capture their non-deterministic relationships, we introduce the probabilistic embedding vector for each channel in MTS as:

$$\boldsymbol{\alpha}_i \sim \mathcal{N}\left(\boldsymbol{\mu}_i, \operatorname{diag}\left(\boldsymbol{\sigma}_i\right)\right) \in \mathbb{R}^d, \boldsymbol{\alpha} = [\boldsymbol{\alpha}_1, ..., \boldsymbol{\alpha}_V]$$

where $\mathcal{N}(\cdot)$ means Gaussian distribution. $\boldsymbol{\alpha}$ refers to the channel embbeding for inputs. To fully take the advantage of uncertainties brought by Gaussian-distributed embeddings, we measure the distance between different channels with the expected likelihood kernel Jebara et al. (2004) as

$$s(\boldsymbol{\alpha}_i, \boldsymbol{\alpha}_j) = \int_{\boldsymbol{x} \in R^n} \mathcal{N}\left(\boldsymbol{x}; \boldsymbol{\mu}_i, \operatorname{diag}(\boldsymbol{\sigma}_i)\right) \mathcal{N}\left(\boldsymbol{x}; \boldsymbol{\mu}_j, \operatorname{diag}(\boldsymbol{\sigma}_j)\right) dx = \mathcal{N}\left(0; \boldsymbol{\mu}_i - \boldsymbol{\mu}_j, \operatorname{diag}(\boldsymbol{\sigma}_i + \boldsymbol{\sigma}_j)\right)$$

Clearly, as a symmetric similarity function, $s(\cdot)$ incorporates both the means and covariances of two Gaussians, considering the uncertainties within them. The parameters of channel embeddings are initialized randomly and then trained along with the rest of the model.

## 2.2 ADAPTIVE GRAPH TRANSFORMER

To consider both temporal and cross-channel dependencies, we introduce graph structure into the Transformer framework and develop an G-Trans module. The detailed structure of G-Trans is shown in Fig. 2 (c). Specifically, we first introduce a **M**ulti-head **S**elf **A**ttention (MSA) block to capture the temporal dependence as

$$\boldsymbol{O} = \operatorname{MSA}(\boldsymbol{X}) = \operatorname{con}\left(\boldsymbol{H}_1, \ldots, \boldsymbol{H}_m\right), \boldsymbol{H}_i = \operatorname{SA}\left(\boldsymbol{Q}_i, \boldsymbol{K}_i, \boldsymbol{V}_i\right) = \operatorname{softmax}(\frac{\boldsymbol{Q}_i^T \boldsymbol{K}_i}{\sqrt{d_K}})\boldsymbol{V}_i \quad (1)$$

where $\boldsymbol{X} \in \mathbb{R}^{V \times T}$ denotes the input and $\operatorname{con}(\cdot)$ means concat operation, $\boldsymbol{Q}_i = \boldsymbol{W}_Q^i \boldsymbol{X} \in \mathbb{R}^{d_k \times T}, \boldsymbol{K}_i = \boldsymbol{W}_K^i \boldsymbol{X} \in \mathbb{R}^{d_k \times T}$, where $i \in \{1, 2, ..., m\}$ and $m$ is the number of heads. It is worth noting that we assign $\boldsymbol{V}_i = \boldsymbol{X}$ to keep the meaning of channels within MTS unchanged for capturing their relationships follow-up. $\boldsymbol{O} \in \mathbb{R}^{(V \times T \times m)}$ is the output of MSA block. Then, different from traditional Transformer that uses feed forward network after multi-head self-attention block, which can not well capture the diverse relationships between different channels, we introduce an **A**daptive **G**raph **C**onvolutional **N**etwork (AGCN) block. Specifically, with the $\boldsymbol{O}$ from MSA block and channel embeddings $\boldsymbol{\alpha}$ defined in Section 2.1, AGCN discoveries channel dependencies automatically as

$$\boldsymbol{A} = \log(s(\boldsymbol{\alpha}, \boldsymbol{\alpha})), \boldsymbol{H} = \ln(\operatorname{softplus}(\operatorname{Conv}(\boldsymbol{O}))), \tilde{\boldsymbol{A}} = \operatorname{softmax}(ReLU(\boldsymbol{A})),$$
$$\tilde{\boldsymbol{h}} = \operatorname{AGCN}(\boldsymbol{\alpha}, \boldsymbol{O}) = \ln(1 + \exp(\boldsymbol{W}\tilde{\boldsymbol{A}}\boldsymbol{H}))] \quad (2)$$

where $\boldsymbol{A} \in \mathbb{R}^{V \times V}$ is the relational matrix calculated by symmetric similarity function and $\boldsymbol{\alpha}$ is updated to be adaptive to the MTS data. $\text{Conv}(\cdot)$ means the convolutional operation, which is utilized to summarize the multi-head information within $\boldsymbol{O}$ into $\boldsymbol{H} \in \mathbb{R}^{V \times T}$. $\tilde{\boldsymbol{A}}$ is the normalized symmetric adjacent matrix. $\boldsymbol{W} \in \mathbb{R}^{K' \times V}$ is the GCN filter. After combining temporal and channel-wise relational information of MTS into $\tilde{\boldsymbol{h}} \in \mathbb{R}^{K' \times T}$, multi-head self attention and feed forward network blocks are further applied for exploring expressive representations of MTS, as the red box shown in Fig. 2 (c), and getting the dynamic latent states $\boldsymbol{h} \in \mathbb{R}^{K \times T}$. Then, we can select a reconstruction or forecasting decoder for different tasks.

## 2.3 VARIATIONAL ADAPTIVE GRAPH TRANSFORMER

Previous Transformer based MTS modeling methods are always equipped with deterministic generative model, which ignores the stochasticity within MTS and has difficulty in modeling complex MTS with sophisticated distribution. To address this issue, we further develop VG-Trans, which is a novel VAE-structured dynamic model equipped with a powerful **E**mbedding guided **P**robabilistic generative **M**odule (EPM) as decoder and G-Trans as encoder. The graphical illustration of the generation (decoder) and inference (encoder) operations of VG-Trans are shown in Fig. 2 (b).

**Embedding guided probabilistic generation:** To consider the stochasticity within MTS, we first introduce a Gaussian-distributed latent variable $\boldsymbol{z}_{t,n} \in \mathbb{R}^K$ at each timestep, and then define the generative process as

$$
\begin{aligned}
\boldsymbol{z}_{t,n} &\sim \mathcal{N}\left(\boldsymbol{\mu}_{t,n}, \text{diag}\left(\boldsymbol{\sigma}_{t,n}\right)\right), \boldsymbol{\mu}_{t,n} = f\left(\boldsymbol{W}_{h,\mu} \boldsymbol{h}_{t-1,n}\right), \\
\boldsymbol{x}_{t,n} &\sim \mathcal{N}\left(\boldsymbol{\mu}_{t,n}^x, \text{diag}\left(\boldsymbol{\sigma}_{t,n}^x\right)\right), \boldsymbol{\mu}_{t,n}^x = f\left(\boldsymbol{W}_{z\mu}^x \boldsymbol{z}_{t,n} + \boldsymbol{W}_{h\mu}^x \boldsymbol{h}_{t-1,n}\right)
\end{aligned}
\tag{3}
$$

As illustrated in Fig. 2 (I) (a) and (b). $\boldsymbol{\mu}_{t,n}$ and $\boldsymbol{\sigma}_{t,n}$ are means and covariance parameters of $\boldsymbol{z}_{t,n}$. $\boldsymbol{h}_{t-1,n} \in \mathbb{R}^{K'}$ denotes the deterministic latent states of G-Trans module. We combine $\boldsymbol{z}_{t,n}$ and $\boldsymbol{h}_{t,n}$ into generative process to consider the temporal dependencies and the stochasticity. $f(\cdot)$ refers to the non-linear activation function. Moreover, to further capture the cross-channel dependencies of inputs and latent variables, we introduce channel embeddings into our generation process by defining

$$
\boldsymbol{W}_{z\mu}^x = \log\left(s\left(\boldsymbol{\alpha}\boldsymbol{\beta}_z\right)\right), \boldsymbol{W}_{h\mu}^x = \text{softmax}\left(\boldsymbol{\alpha}\boldsymbol{\beta}_h\right)
\tag{4}
$$

The channel embeddings are incorporated into generative process by defining the factor loading matrices $\boldsymbol{W}_{z\mu}^x$ and $\boldsymbol{W}_{h\mu}^x$ as the mapping function of them, which can capture the non-deterministic inter-relationships between channels, as introduced in Dieng et al. (2020), thus to improve the capacity of model in modeling complex MTS. $\boldsymbol{\beta}_z \in \mathbb{R}^{d \times K}$ is the mapping matrix that transmit $\boldsymbol{z}_{t,n}$ into the embedding space of $\boldsymbol{x}_{t,n}$, while $\boldsymbol{\beta}_h \in \mathbb{R}^{d \times K'}$ is the mapping matrix that transmit $\boldsymbol{h}_{t,n}$ into the embedding space of $\boldsymbol{x}_{t,n}$. We call our generative model as EPM. Compared with generative module of previous Transformer based methods for MTS modeling, EPM discovers the latent semantic structure of each channel as an probabilistic embedding vector and capturing the relationships between each other according to the similarity of channel embeddings, meanwhile, considering the non-deterministic within them, thus to improve the representative capacity.

**Inference:** The purpose of inference module is to map the inputs $\boldsymbol{x}_n$ to $\boldsymbol{z}_n$. To consider the locality temporal dependencies, we first apply a convolutional operation on $\boldsymbol{x}_{t,n}$ as $\hat{\boldsymbol{x}}_{t,n} = \text{Conv}(\boldsymbol{x}_{t,n}) \in \mathbb{R}^V$. Then, with MSA and AGCN blocks, we summarize temporal and channel-wise relational information of input MTS as

$$
\boldsymbol{O}_n = \text{MSA}(\hat{\boldsymbol{x}}_n), \tilde{\boldsymbol{h}}_n = \text{AGCN}\left(\boldsymbol{O}_n, \boldsymbol{\alpha}\right)
\tag{5}
$$

Given time series $\tilde{\boldsymbol{h}}_{1:T,n}$, we apply a linear projection and combine it with a positional embedding to obtain

$$
\boldsymbol{h}_{t,n}^{(0)} = \text{LayerNorm}(\text{MLP}(\tilde{\boldsymbol{h}}_{t,n}) + \text{Position}(t))
\tag{6}
$$

With $\boldsymbol{h}_{t,n}^{(0)}$ as the input, we then apply multi-head self attention and feed forward network blocks to get the dynamic latent states $\boldsymbol{h}_t$, as shown in Fig. 2 (c). Following VAE based models Kingma & Welling (2014), we define a Gaussian distributed variational distribution $q(\boldsymbol{z}_{t,n}) = \mathcal{N}(\boldsymbol{\mu}_{t,n}, \text{diag}(\boldsymbol{\sigma}_{t,n}))$ to approximate the true posterior distribution $p(\boldsymbol{z}_{t,n}|-)$, and map the dynamic layent states $\boldsymbol{h}_t$ to their parameters as:

$$
\boldsymbol{\mu}_{t,n} = f\left(\boldsymbol{C}_{x\mu} \boldsymbol{h}_{t,n} + \boldsymbol{b}_{x\mu}\right), \boldsymbol{\sigma}_{t,n} = \text{Softplus}\left(f\left(\boldsymbol{C}_{x\sigma} \boldsymbol{h}_{t,n} + \boldsymbol{b}_{x\sigma}\right)\right)
\tag{7}
$$

where $\boldsymbol{C}_{x\mu}, \boldsymbol{C}_{x\sigma} \in \mathbb{R}^{K \times V}$, $\boldsymbol{b}_{x\mu}, \boldsymbol{b}_{x\sigma} \in \mathbb{R}^K$ are all learnable parameters of the inference network.

## 3 MODEL TRAINING

As mentioned in Cao et al. (2020); Wen et al. (2022), the prediction-based model is expert in capturing the periodic information of the MTS, while the reconstruction-based model can explore the global distribution of the time series. To combine the complementary advantages of the two models for facilitating the representation capability of MTS, we formulate the optimization function as the combination of both prediction and reconstruction losses and define the marginal likelihood as

$$P(\mathcal{D}|\boldsymbol{\alpha}, \boldsymbol{W}) = \int \prod_{t=1}^{T} p(\boldsymbol{x}_{t,n} \mid \boldsymbol{z}_{t,n}, \boldsymbol{\alpha}) + p(\boldsymbol{x}_{T,n} \mid \boldsymbol{h}_{1:T-1,n}, \boldsymbol{\alpha}) d\boldsymbol{z}_{1:T,n}$$

where the first and the second term are reconstruction and prediction loss separately. Similar to VAEs, with the inference network and variational distribution in Eq. equation 7, the optimization objective of VG-Trans can be achieved by maximizing the evidence lower bound (ELBO) of the log marginal likelihood, which can be computed as

$$\mathcal{L} = \sum_{n=1}^{N} \left[ \sum_{t=1}^{T} \mathbb{E}_{q(\boldsymbol{z}_{t,n})} [\ln p\left(\boldsymbol{x}_{t,n} \mid \boldsymbol{z}_{t,n}\right) + \gamma \left[\ln p\left(\boldsymbol{x}_{T,n} \mid \boldsymbol{h}_{1:T-1,n}\right)\right] - \ln \frac{q\left(\boldsymbol{z}_{t,n}|\boldsymbol{x}_{t,n}\right)}{p(\boldsymbol{z}_{t,n}|\boldsymbol{h}_{t-1,n})}]] \quad (8)$$

where $\gamma > 0$ is a hyper-parameter to balance the prediction and the reconstruction losses, which is chosen by grid search on the validation set. The parameters of channel embedding $\boldsymbol{\alpha}$ can be learned with Bayes by Backprop (Blundell et al., 2015) as it can be reparameterized as $s(\boldsymbol{\alpha}_i, \boldsymbol{\alpha}_j) = (\boldsymbol{\mu}_i - \boldsymbol{\mu}_j) + (\text{diag}(\boldsymbol{\sigma}_i + \boldsymbol{\sigma}_j)) * \boldsymbol{\epsilon}_{ij}$. The detailed procedures of the optimization of VG-Trans are summarized in Appendix.

## 4 APPLICATION TO ANOMALY DETECTION TASK

Anomaly detection of MTS is defined as a problem that determines whether an observation from a certain task and at a certain time is anomalous or not. Specifically, the model is trained to learn normal patterns of multivariate time series, the more an observation follows normal patterns, the more likely it can be reconstructed and predicted well with higher confidence. Our model is an unsupervised probabilistic generative model, which can be applied to unsupervised anomaly detection directly. Specifically, we apply the reconstruction and prediction error of $\boldsymbol{x}_t$ as the anomaly score to determine whether an observed variable is anomalous or not, and it is computed as

$$\mathcal{S}_{t,n} = \left(\mathcal{S}_{t,n}^{r} + \gamma(-\mathcal{S}_{t,n}^{p})\right)/(1+\gamma), \mathcal{S}_{t,n}^{r} = \log p(\boldsymbol{x}_{t,n}|\boldsymbol{z}_{t,n}), \mathcal{S}_{t,n}^{p} = (\boldsymbol{x}_{t,n} - \hat{\boldsymbol{x}}_{t,n})^2 \quad (9)$$

where $\mathcal{S}_{t,n}^{r}$ and $\mathcal{S}_{t,n}^{p}$ are reconstruction and prediction score, respectively. An observation $x_t$ will be classified as anomalous when $\mathcal{S}_{t,n}$ is below a specific threshold. From a practical point of view, we use the Peaks-Over-Threshold (POT) (Siffer et al., 2017) approach to help select threshold.

Moreover, we note that after collecting both non-deterministic temporal and channel dependencies information within VG-Trans and get latent representations of MTS, we can introduce more powerful structure for different specific tasks. Specifically, we combine our model with Anomaly Transformer (Xu et al., 2022) by introducing anomaly attention and association discrepancy into VG-Trans and get the VG-Anomaly-Trans.

## 5 APPLICATION TO FORECASTING TASK

Considering forecasting task of MTS being formulated as finding the function to predict the next $\tau$ time steps (Bai et al., 2020) given the past $T$ time steps as

$$\{\boldsymbol{x}_{:,t+1}, \boldsymbol{x}_{:,t+2}, \dots, \boldsymbol{x}_{:,t+\tau}\} = \mathcal{F}_\theta\left(\boldsymbol{x}_{:,t}, \boldsymbol{x}_{:,t-1}, \dots, \boldsymbol{x}_{:,t-T+1}\right) \quad (10)$$

where $\boldsymbol{\theta}$ refers to the learnable parameter. Since VG-Trans is a representation learning method, we can refer to it as a play and plug framework, which can be applied to forecasting task by combining with corresponding methods. Inspired by the effectiveness of Autoformer in forecasting, we combine it with our proposed VG-Trans and get the VG-Autoformer. As shown in Fig. 3, VG-Autoformer first get latent representations of MTS

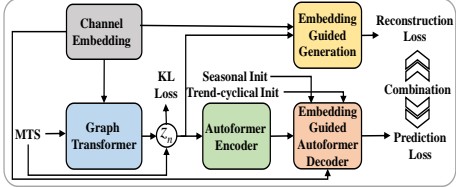

Figure 3: The framework of VG-Autoformer.

Table 1: F1-score performance of different methods.

| Methods | Size | CDN | | | SMD | | | MSL | | | SMAP | | |
|---|---|---|---|---|---|---|---|---|---|---|---|---|---|
| | | P | R | F1_score | P | R | F1_score | P | R | F1_score | P | R | F1_score |
| OCSVM | 10 | 0.508 | 0.640 | 0.566 | 0.443 | 0.767 | 0.562 | 0.598 | 0.868 | 0.708 | 0.538 | 0.591 | 0.563 |
| IsolationForst | 10 | 0.496 | 0.640 | 0.559 | 0.423 | 0.733 | 0.536 | 0.539 | 0.865 | 0.664 | 0.524 | 0.590 | 0.555 |
| LOF | 10 | 0.667 | 0.364 | 0.671 | 0.563 | 0.399 | 0.467 | 0.477 | 0.853 | 0.612 | 0.589 | 0.563 | 0.576 |
| LSTM | 10 | 0.585 | 0.614 | 0.599 | 0.786 | 0.852 | 0.818 | 0.855 | 0.825 | 0.840 | 0.894 | 0.781 | 0.834 |
| VRNN | 10 | 0.957 | 0.901 | 0.928 | 0.970 | 0.795 | 0.874 | 0.884 | 0.902 | 0.893 | 0.805 | 0.821 | 0.813 |
| DOMI | 10 | 0.938 | 0.878 | 0.907 | 0.943 | 0.913 | 0.927 | 0.896 | 0.639 | 0.746 | 0.864 | 0.567 | 0.685 |
| BeatGAN | 10 | 0.921 | 0.887 | 0.904 | 0.729 | 0.841 | 0.781 | 0.898 | 0.854 | 0.875 | 0.924 | 0.559 | 0.696 |
| OmniAnomaly | 10 | 0.983 | 0.875 | 0.926 | 0.837 | 0.868 | 0.852 | 0.890 | 0.864 | 0.877 | 0.925 | 0.820 | 0.869 |
| SDFVAE | 10 | 0.972 | 0.903 | 0.936 | 0.882 | 0.926 | 0.903 | 0.853 | 0.894 | 0.873 | 0.884 | 0.908 | 0.896 |
| GNN | 10 | 0.933 | 0.902 | 0.917 | 0.829 | 0.964 | 0.891 | 0.881 | 0.889 | 0.885 | 0.812 | 0.944 | 0.873 |
| Interfusion | 10 | 0.980 | 0.883 | 0.929 | 0.870 | 0.854 | 0.862 | 0.813 | 0.927 | 0.866 | 0.898 | 0.885 | 0.891 |
| THOC | 10 | 0.943 | 0.902 | 0.922 | 0.798 | 0.909 | 0.850 | 0.884 | 0.910 | 0.897 | 0.921 | 0.893 | 0.907 |
| Trans | 10 | 0.861 | 0.802 | 0.830 | 0.853 | 0.748 | 0.797 | 0.856 | 0.813 | 0.834 | 0.913 | 0.568 | 0.700 |
| Anomaly-Trans | 10 | 0.888 | 0.991 | 0.937 | 0.894 | 0.954 | 0.923 | 0.921 | 0.951 | 0.936 | 0.941 | 0.994 | 0.967 |
| VG-Trans | 10 | 0.903 | 0.998 | 0.948 | 0.972 | 0.895 | 0.932 | 0.930 | 0.959 | 0.944 | 0.984 | 0.928 | 0.955 |
| VG-Anomaly-Trans | 10 | 0.948 | 0.953 | **0.950** | 0.925 | 0.933 | **0.938** | 0.934 | 0.964 | **0.949** | 0.974 | 0.964 | **0.969** |

Figure 4: Visualization of example channels and corresponding relational matrices of $\boldsymbol{x}_n$ on SMD machine-2-3 dataset.

with VG-Trans, then transport them with Autoformer encoder. Finally, after by combining channel embedding with Autoformer decoder, we develop an embedding guided Autoformer decoder as:

$$\boldsymbol{x}_{T+1:T+\tau} = \boldsymbol{W}_{seasonal}\boldsymbol{o}_{T+1:T+\tau}^{seasonal} + \boldsymbol{W}_{trend}\boldsymbol{o}_{T+1:T+\tau}^{trend}, \quad \boldsymbol{W}_{trend} = softmax(\boldsymbol{\alpha}\boldsymbol{\beta}_x) \quad (11)$$

where $\boldsymbol{o}_{T+1:T+\tau}^{seasonal} \in \mathbb{R}^{V\times\tau}$ and $\boldsymbol{o}_{T+1:T+\tau}^{trend} \in \mathbb{R}^{V\times\tau}$ separately represent the seasonal and trend-cyclical outputs of Autoformer decoder, which is refined by channel embedding $\boldsymbol{\alpha}$ via a learnable parameter $\boldsymbol{\beta}_x \in \mathbb{R}^{d\times V}$. The intuition behind this is that long-term channel relationships may lie in trend-cyclical part rather than seasonal part. The detailed structure of VG-Autoformer is introduced in Appendix. To optimize VG-Autoformer for forecasting, we modify the loss in Eq. 12 as

$$\mathcal{L} = \sum_{n=1}^{N}[\sum_{t=1}^{T}\mathbb{E}_{q(\boldsymbol{z}_{t,n})}[\ln p(\boldsymbol{x}_{t,n} \mid \boldsymbol{z}_{t,n}) + \gamma\ln p(\boldsymbol{x}_{T+1:T+\tau,n} \mid \boldsymbol{h}_{1:T,n}) - \ln\frac{q(\boldsymbol{z}_{t,n} \mid \boldsymbol{x}_{t,n})}{p(\boldsymbol{z}_{t,n} \mid \boldsymbol{h}_{t-1,n})}]]$$

## 6 EXPERIMENT

We conduct extensive experiments to evaluate the performance of our proposed models on forecasting and anomaly detection tasks of MTS.

**Datasets and set up:** We evaluate the effectiveness of our model on twofold datasets:1) four real-world datasets for anomaly detection, including CDN Dai et al. (2021), SMD, MSL and SMAP Xu et al. (2022); 2) four datasets for forecasting, including ETTh, ETTm, Weather and ECL (Zhou et al., 2021). The results are either quoted from the original papers or reproduced with the code provided by the authors. The way of data preprocessing is the same as Dai et al. (2021), where the window size $T$ and overlap $o$ are set as $T = 20, o = 5$ for anomaly detection and $T = 20, o = 0$ for forecasting. Adam optimizer Kingma & Ba (2015) is employed with learning rate of 0.0002, the batch size is set to be 64. The number of heads in VG-Trans is set as 8. The summary statistics of these datasets and other implementation details are described in Appendix.

### 6.1 ANOMALY DETECTION

Similar to the previous studies (Dai et al., 2021), we employ Precision, Recall, and F1-score as the evaluation metrics to indicate the anomaly detection performance of different methods. Particularly,

Table 2: Multivariate results with predicted length as {96, 168, 288, 336} on the five datasets and {24,48,96,168} on the Weather dataset, lower scores are better. Metrics are averaged over 5 runs, best results are highlighted in bold.

| Models Metric | | VG-autoformer | | Autoformer | | Informer | | LogTrans | | Reformer | | LSTNet | |
|---|---|---|---|---|---|---|---|---|---|---|---|---|---|
| | | MSE | MAE | MSE | MAE | MSE | MAE | MSE | MAE | MSE | MAE | MSE | MAE |
| ETTh1 | 96 | **0.420** | **0.435** | 0.426 | 0.442 | 0.925 | 0.761 | 1.008 | 0.875 | 0.774 | 0.637 | 1.257 | 0.983 |
| | 168 | **0.458** | **0.457** | 0.490 | 0.481 | 0.931 | 0.752 | 1.002 | 0.846 | 1.824 | 1.138 | 1.997 | 1.214 |
| | 288 | **0.502** | **0.483** | 0.514 | 0.494 | 1.120 | 0.835 | 1.141 | 0.972 | 0.910 | 0.700 | 1.528 | 1.376 |
| | 336 | **0.494** | **0.482** | 0.505 | 0.484 | 1.128 | 0.873 | 1.362 | 0.952 | 2.177 | 1.280 | 2.655 | 1.369 |
| ETTh2 | 96 | **0.383** | 0.415 | 0.380 | **0.413** | 2.975 | 1.359 | 3.129 | 1.297 | 1.656 | 1.054 | 3.367 | 1.982 |
| | 168 | **0.429** | **0.443** | 0.457 | 0.455 | 3.489 | 1.515 | 4.070 | 1.681 | 4.660 | 1.846 | 3.242 | 2.513 |
| | 288 | **0.459** | **0.463** | 0.470 | 0.468 | 6.020 | 2.030 | 7.172 | 2.471 | 2.730 | 1.346 | 8.252 | 2.665 |
| | 336 | **0.458** | **0.467** | 0.471 | 0.475 | 2.723 | 1.340 | 3.875 | 1.763 | 4.028 | 1.688 | 2.544 | 2.591 |
| ETTm1 | 96 | 0.484 | 0.465 | **0.481** | **0.463** | 0.678 | 0.614 | 0.768 | 0.792 | 1.433 | 0.945 | 2.762 | 1.542 |
| | 168 | **0.505** | **0.477** | 0.573 | 0.506 | 0.748 | 0.634 | 0.886 | 0.759 | 0.915 | 0.696 | 1.124 | 0.897 |
| | 288 | **0.520** | **0.492** | 0.634 | 0.528 | 1.056 | 0.789 | 1.462 | 1.320 | 1.820 | 1.094 | 1.257 | 2.076 |
| | 336 | **0.536** | **0.479** | 0.541 | 0.503 | 1.043 | 0.759 | 1.412 | 1.125 | 1.004 | 0.731 | 1.786 | 1.598 |
| ETTm2 | 96 | **0.246** | **0.320** | 0.255 | 0.339 | 0.365 | 0.453 | 0.768 | 0.642 | 0.658 | 0.619 | 3.142 | 1.365 |
| | 168 | **0.259** | **0.322** | 0.274 | 0.338 | 0.681 | 0.647 | 1.291 | 1.173 | 1.255 | 0.858 | 1.502 | 1.328 |
| | 288 | **0.328** | **0.365** | 0.342 | 0.378 | 1.047 | 0.804 | 1.090 | 0.806 | 2.441 | 1.190 | 2.856 | 1.329 |
| | 336 | **0.325** | **0.370** | 0.339 | 0.372 | 1.438 | 0.921 | 2.006 | 1.334 | 2.213 | 1.104 | 3.259 | 1.576 |
| Weather | 24 | **0.156** | **0.237** | 0.182 | 0.265 | 0.213 | 0.287 | 0.365 | 0.405 | 0.176 | 0.248 | 0.575 | 0.507 |
| | 48 | **0.220** | **0.264** | 0.228 | 0.306 | 0.335 | 0.387 | 0.496 | 0.485 | 0.284 | 0.344 | 0.622 | 0.553 |
| | 96 | **0.220** | **0.273** | 0.266 | 0.336 | 0.300 | 0.384 | 0.458 | 0.490 | 0.689 | 0.596 | 0.594 | 0.587 |
| | 168 | **0.263** | **0.327** | 0.319 | 0.376 | 0.429 | 0.465 | 0.649 | 0.573 | 0.423 | 0.451 | 0.676 | 0.585 |
| ECL | 96 | **0.193** | **0.307** | 0.201 | 0.317 | 0.274 | 0.368 | 0.258 | 0.357 | 0.312 | 0.402 | 0.680 | 0.645 |
| | 168 | **0.202** | **0.313** | 0.232 | 0.341 | 0.278 | 0.377 | 0.290 | 0.382 | 0.318 | 0.413 | 0.318 | 0.368 |
| | 288 | **0.231** | **0.339** | 0.234 | 0.343 | 0.315 | 0.410 | 0.319 | 0.425 | 0.353 | 0.437 | 0.481 | 0.612 |
| | 336 | **0.266** | **0.358** | 0.270 | 0.361 | 0.300 | 0.394 | 0.280 | 0.380 | 0.350 | 0.433 | 0.828 | 0.727 |

F1-score is deemed as a comprehensive indicator since it balances precision and recall. We extensively compare our model with 14 baselines,including the classic methods: OC-SVM (Tax & Duin, 2004), IsolationForest (Liu et al., 2008), LOF (Breunig et al., 2000); recurrent structure methods: LSTM (Hundman et al., 2018), THOC (Shen et al., 2020); probabilistic dynamic models: LSTM-VAE (Park et al., 2018) VRNN (Chung et al., 2015), DOMI (Su et al., 2021), BeatGAN (Zhou et al., 2019), OmniAnomaly (Su et al., 2019), SDFVAE (Dai et al., 2021); methods considering cross-channel dependency: Interfusion (Li et al., 2021), GNN (Deng & Hooi, 2021); and Transformer based methods: Transformer (Zerveas et al., 2021), Anomaly-Trans (Xu et al., 2022). Anomaly-Trans is the state-of-the-art method based on Transformer structure.

Firstly, we compare the proposed models with baselines on their detection performance and report the average F1-score results on five independent runs in Table.1, the best results are highlighted in boldface. It is obvious that deep learning based methods outperform the classic methods for stronger representational power. Probabilistic dynamic methods achieve better results than LSTM, since they consider the stochasticity within MTS. Both recurrent and graph structures can boost the performance, indicating the effectiveness of temporal and cross-channel relationships on learning normal patterns of MTS. Original Transformer is not suitable for MTS modeling, leading to a poor performance, while Anomaly-Trans achieves SOTA results before our models, showing the efficient of Transformer structure in modeling long and complex temporal dependencies. Our proposed VG-Trans achieves the best detection performance among all methods on most datasets, which demonstrates its effectiveness in modeling non-deterministic temporal and cross-channel dependencies, thus to learn more expressive representations of the normal pattern of MTS. Finally, the performance of VG-trans can be further improved by combining it with Anomaly-Trans for VG-Anomaly-Trans.

To better demonstrate the effectiveness of capturing the channel-wise relationships of VG-Trans, we further visualize some channels of $x_n$ in Fig. 4, and presents a subset of the corresponding relational matrices to these channels in Fig. 4 (middle). As we can see, relational matrices can effectively reflect the similarity and correlation between channels, such as channels $21, 23, 25$ of $x_n$, which illustrates the capacity of VG-Trans in capturing the cross-channel dependencies within MTS.

## 6.2 FORECASTING

We deploy two widely used metrics, Mean Absolute Error (MAE) and Mean Square Error (MSE) (Zhou et al., 2021), to measure the performance of forecasting models. Six popular methods are compared here, including recurrent structure models: LSTnet (Lai et al., 2018), LSTMa (Bahdanau

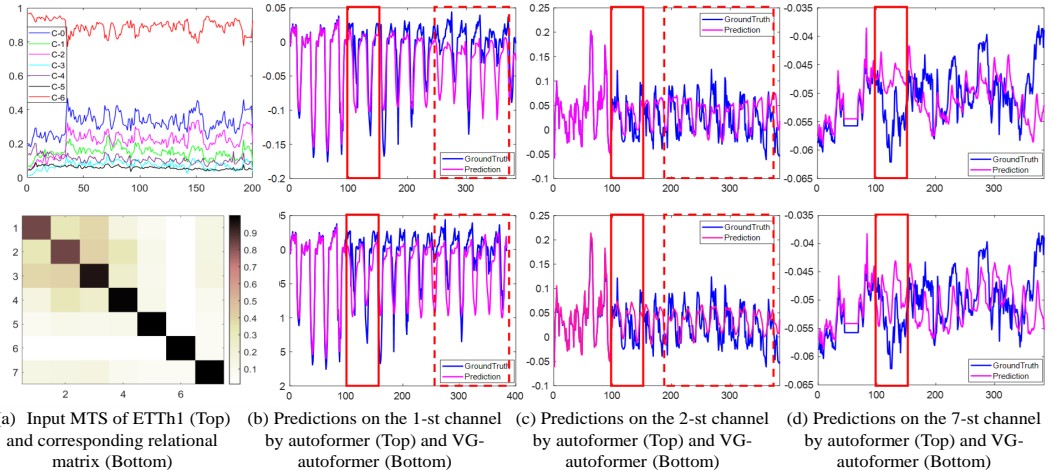

(a) Input MTS of ETTh1 (Top) and corresponding relational matrix (Bottom)

(b) Predictions on the 1-st channel by autoformer (Top) and VG-autoformer (Bottom)

(c) Predictions on the 2-st channel by autoformer (Top) and VG-autoformer (Bottom)

(d) Predictions on the 7-st channel by autoformer (Top) and VG-autoformer (Bottom)

Figure 5: Visualizations on ETTh1 dataset of (a) the input MTS under the input-200 setting and the learned relational matrix among different channels; (b), (c), (d): predictions on different channels by autoformer and VG-autoformer.

et al., 2015); graph structure model: AGCRN (Bai et al., 2020); Transformer based models: Reformer (Kitaev et al., 2019), LogTrans (Li et al., 2019), Informer (Zhou et al., 2021) and Autoformer (Kingma & Welling, 2014). We note that the experiment settings used here are same with Zhou et al. (2021).

Table 2 presents the overall prediction performance, which are average MAE and MSE on five independent runs, and the best results are highlighted in boldface. As we can see, RNN based methods underperform Transformer based methods on most setting, especially on long-term forecasting, suggesting the efficiency of Transformer structure in modeling complex and long-distance temporal dependencies. By incorporating graph convolutional structure into network, AGCRN outperforms other RNN based methods, demonstrating the importance of modeling cross-channel dependency explicitly. Combining VG-Trans and autoformer, VG-Autoformer achieves much better results

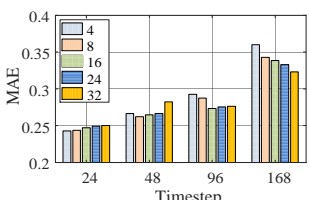

Figure 6: The influence of embedding size on forecasting.

on almost all datasets, suggesting efficient of VG-Trans in capturing the non-deterministic complex temporal dependencies and complex cross-channel relationships in MTS, which helps to get robust representations, thus ensuring the promising predictions. Besides, we also test the effect of embedding size on forecasting performance with Weather dataset, as shown in Fig. 6, we set $d = 4, 8, 16, 24, 32$ and test the MAE of VG-Trans with different $d$. We can find that the selection of embedding size is affected by the length of horizons, higher embedding size performs better on longer horizons for more complex channel-wise relationships.

In addition to quantitative evaluation, we also list the case study of prediction results by autoformer and VG-autoformer in Fig 5. As shown in Fig 5 (a), similar with anomaly detection task, relational matrices in forecasting task can also effectively reflect the similarity and correlation between channels, such as channels 1,2 and 3. By the aid of the relationships between different channels captured by VG-autoformer, the prediction results of the related channels can guide and revise each other, thus to ensure the forecasting performance. As illustrated in red boxes of Fig 5 (b), (c) and (d), VG-autoformer achieves more accurate prediction than autoformer.

### 6.3 ABLATION STUDY

We conduct ablation study to analyze the importance of each component in our model, including graph structure, variational scheme, embedding guided generation and the combined optimization objective. The results on both anomaly detection and forecasting tasks are listed in Table 3. Firstly, on both tasks, all the structural components contribute to the performance of the framework. Specifically, by introducing a powerful variational generative module for Transformer or graph Transformer, their performance gain a significant improvement for being able to get robust representation of MTS with complex distributions. Meanwhile, as shown in the last two lines, incorporating cross-channel



Figure 7: Case study of anomaly scores by Transformer (left), G-Trans (middle) and VG-Trans (right) on machine 2-3 of SMD dataset. Regions highlighted in red and blue represent the groundtruth anomaly segments and misjudgement segments by methods, red lines refer to the threshold selected according to the rule that all anomalies can be detected.

Table 3: Ablation study of VG-Trans on anomaly detection and forecasting tasks.

| Architecture | Variational Generation | Embedding Guided Generation | Optimization Objective | Anomaly Detection | | | | Forecasting | | | | | |
|---|---|---|---|---|---|---|---|---|---|---|---|---|---|
| | | | | CDN | SMD | MSL | SMAP | ETTh1 | | ETTh2 | | ETTm1 | |
| | | | | F1-score | F1-score | F1-score | F1-score | 168 | 288 | 168 | 288 | 168 | 288 |
| Transformer | ✗ | ✗ | Prediction | 0.830 | 0.797 | 0.834 | 0.700 | 0.490 | 0.514 | 0.457 | 0.470 | 0.573 | 0.634 |
| | ✓ | ✗ | Reconstruction | 0.911 | 0.907 | 0.911 | 0.922 | - | - | - | - | - | - |
| | ✓ | ✗ | Combined | 0.922 | 0.920 | 0.921 | 0.937 | - | - | - | - | - | - |
| Graph Transformer | ✗ | ✗ | Reconstruction | 0.910 | 0.907 | 0.908 | 0.923 | - | - | - | - | - | - |
| | ✗ | ✗ | Prediction | 0.895 | 0.894 | 0.902 | 0.908 | 0.478 | 0.509 | 0.448 | 0.465 | 0.547 | 0.602 |
| | ✗ | ✗ | Combined | 0.919 | 0.917 | 0.920 | 0.931 | - | - | - | - | - | - |
| | ✓ | ✗ | Combined | 0.940 | 0.928 | 0.935 | 0.949 | 0.470 | 0.506 | 0.441 | 0.463 | 0.537 | 0.589 |
| | ✓ | ✓ | Combined | **0.948** | **0.932** | **0.944** | **0.955** | **0.458** | **0.502** | **0.429** | **0.459** | **0.505** | **0.520** |

dependency modeling into generative process can further improve the generative capacity of models, thus resulting a better performance on two tasks. In addition, by comparing the 1-st and 5-th lines, the 3-rd and 6-th lines, the efficiency of graph structure can be proved. The combined optimization objective is also beneficial to learn expressive representation of MTS as shown in the 5-th and 7-th lines. These verify that each module of our design is effective and necessary.

To further show the efficiency of different components of VG-Trans in capturing the robust representations of MTS intuitively, focusing on anomaly detection task, we visualize the anomaly score of the case study. We compare the anomaly scores by Transformer, G-Trans and VG-Trans, the results are visualized in Fig. 7. As the deterministic methods, Trans and G-Trans get turbulent anomaly scores since they ignore the stochastic of MTS. Considering the inter-relationships within MTS, G-Trans exhibits more distinct spikes in the regions of anomalies, thus leading to less regions misjudged to anomalies. As probabilistic methods, VG-Trans realizes smoother anomaly scores than previous methods for considering stochasticity within MTS. Meanwhile, it exhibits considerable spikes in the regions of anomalies for considering both temporal and cross-channel dependencies. These properties enable that only one region is misjudged by VG-Trans, thus leading to better F1-score. The anomaly score of this case further demonstrates the capability of graph structure and powerful probabilistic generative process in VG-Trans in learning expressive representations of complex MTS, which echoes the numerical results listed in Table 3.

## 7 CONCLUSION

In this paper, towards MTS modeling tasks, we propose a novel variational dynamic model named VG-Trans, which consists of G-Trans module that can capture both cross-channel and long-distance temporal dependencies by incorporating graph into Transformer, and an embedding guided probabilistic generative module to consider the stochasticity within MTS and enhance the capacity of modeling MTS with complex distribution. To achieve efficient optimization, we introduce an autoencoding variational inference scheme with a combined prediction and reconstruction loss. VG-Trans is able to get the robust representations of MTS, which enables it to be combined with the existed methods and applied to different tasks of MTS. Experiments on both anomaly detection and forecasting tasks of MTS illustrate the effectiveness of our model in both extracting expressive representations of MTS.

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

# A APPENDIX

## A.1 ALGORITHM

The autoencoding variational inference algorithm for VG-Trans is described in Algorithm. 1.

It is a great optimization challenge to train a dynamic VAE structured model in practice, due to the well-known posterior collapse and unbounded KL divergence in the objective. Here, we utilize two approaches for stabilizing the training.

**Warm-up:** The variational training criterion in Eq. equation 12 contains the likelihood term $p(\boldsymbol{x}_j|\boldsymbol{\Phi}^{(1)}, \boldsymbol{\theta}_j^{(1)})$ and the variational regularization term. During the early training stage, the variational regularization term causes some of the latent units to become inactive before their learning useful representation **?**. We solve this problem by first training the parameters only using the reconstruction error, and then adding the KL loss gradually with a temperature coefficient:

$$\mathcal{L} = \sum_{n=1}^{N} [\sum_{t=1}^{T} \mathbb{E}_{q(\boldsymbol{z}_{t,n})}[\ln p\left(\boldsymbol{x}_{t,n} \mid \boldsymbol{z}_{t,n}\right) + \gamma\left[\ln p\left(\boldsymbol{x}_{T,n} \mid \boldsymbol{h}_{1:T-1,n}\right)\right] - \beta\ln \frac{q(\boldsymbol{z}_{t,n}|\boldsymbol{x}_{t,n})}{p(\boldsymbol{z}_{t,n}|\boldsymbol{h}_{t-1,n})}]] \quad (12)$$

where $\beta$ is increased from 0 to 1 during the first N training epochs.

**Gradient clipping:** Optimizing the unbounded KL loss often causes the sharp gradient during training, we address this by clipping gradient with a large L2-norm above a certain threshold, which we set 20 in all experiments. This technique can be easily implemented and allows networks to train smoothly.

## A.2 THE DERIVATION OF ELBO

$$P\left(\mathcal{D} \mid \alpha, W\right)$$

$$= \int \prod_{n=1}^{N} \prod_{t=1}^{T} p\left(x_{t,n} \mid z_{t,n}, \alpha\right) * p\left(x_{T,n} \mid h_{1:T-1,n}, \alpha\right) * p\left(z_{t,n} \mid h_{T-1,n}\right) dz_{1:T,n}, \ln P\left(\mathcal{D} \mid \alpha, W\right)$$

$$= \ln \int \prod_{n=1}^{N} \prod_{t=1}^{T} p\left(x_{t,n} \mid z_{t,n}, \alpha\right) p\left(x_{T,n} \mid h_{1:T-1,n}, \alpha\right) p\left(z_{t,n} \mid h_{T-1,n}\right) dz_{1:T,n}$$

$$= \ln \int \prod_{t=1}^{T} \prod_{t=1}^{T} p\left(x_{t,n} \mid z_{t,n}\right) p\left(x_{T,n} \mid h_{1:T-1,n}\right) p\left(z_{t,n} \mid h_{T-1,n}\right) \frac{q(z_{t,n} \mid x_{t,n})}{q(z_{t,n} \mid x_{t,n})} dz_{1:T,n} \quad (13)$$

$$= \ln \int \prod_{t=1}^{T} \prod_{t=1}^{T} \mathbb{E}_{q(z_{t,n})} \left[ p\left(x_{t,n} \mid z_{t,n}\right) p\left(x_{T,n} \mid h_{1:T-1,n}\right) \frac{p\left(z_{t,n} \mid h_{T-1,n}\right)}{q\left(z_{t,n} \mid x_{t,n}\right)} \right]$$

$$\geq \sum_{n=1}^{N} \left[ \sum_{t=1}^{T} \mathbb{E}_{q(z_{t,n})} \left[ \ln p\left(x_{t,n} \mid z_{t,n}\right) + \gamma\left[\ln p\left(x_{T,n} \mid h_{1:T-1,n}\right)\right] + \ln \frac{p\left(z_{t,n} \mid h_{t-1,n}\right)}{q\left(z_{t,n} \mid x_{t,n}\right)} \right] \right]$$

$$= \sum_{n=1}^{N} \left[ \sum_{t=1}^{T} \mathbb{E}_{q(z_{t,n})} \left[ \ln p\left(x_{t,n} \mid z_{t,n}\right) + \gamma\left[\ln p\left(x_{T,n} \mid h_{1:T-1,n}\right)\right] - \ln \frac{q\left(z_{t,n} \mid x_{t,n}\right)}{p\left(z_{t,n} \mid h_{t-1,n}\right)} \right] \right]$$

## A.3 THE NOTATION TABLE OF OUR PAPER

To better understand the proposed model, we summarize the notations used in this paper in Table 4.

## A.4 DATASETS

**Anomaly detection:** For anomaly detection task, we conduct extensive experiments on four datasets: one real-world dataset named CDN multivariate KPI dataset and three public datasets named SMD, MSL and SMAP that were released by the works Su et al. (2019) and Hundman et al. (2018), respectively. The basic statistical information of datasets is reported in Table 5 and Table 6. CDN multivariate KPIs dataset is collected from a large internet company in China, and the dataset contains 12 websites monitored with 36 KPIs individually. These websites are different from each other in

---

**Algorithm 1** Autoencoding Variational Inference of VG-Trans

---

Set mini-batch size as $M$, the number of convolutional filters $K$ and hyperparameters;
Initialize the parameters of inference networks $\boldsymbol{\Omega}$, EPM $\boldsymbol{\Psi}$, G-Trans $\boldsymbol{\theta}$ and the channel embeddings $\boldsymbol{\alpha}^{(0:1)}$;
**repeat**
    Randomly select a mini-batch of $M$ MTS consist of $T$ subsequences to form a subset $\{\boldsymbol{x}_{1:T,i}\}_{i=1}^{M}$;
    Draw random noise $\{\boldsymbol{\epsilon}_{t,n}\}_{t=1,n=1}^{T,M}$, $\{\boldsymbol{\epsilon}_i^{(0)}\}_{i=1}^{V}$, from uniform distribution for sampling latent states $\{\boldsymbol{z}_{t,n}\}_{t=1,n=1}^{T,M}$ and channel embbedings $\{\boldsymbol{\alpha}_i\}_{i=1}^{V}$;
    Calculate $\nabla\mathcal{L}\left(\boldsymbol{\Omega},\boldsymbol{\Psi};X,\boldsymbol{\epsilon}_{t,n},\boldsymbol{\epsilon}_i,\boldsymbol{\theta},\boldsymbol{\alpha}\right)$ according to Eq. (8), and update parameters of inference module $\boldsymbol{\Omega}$, EPM $\boldsymbol{\Psi}$, G-trans module $\boldsymbol{\theta}$, as well as the channel embeddings $\boldsymbol{\alpha}$ jointly;
**until** convergence
return global parameters $\{\boldsymbol{\Omega},\boldsymbol{\Psi},\boldsymbol{\theta},\boldsymbol{\alpha}\}$.

---

Table 4: Notation table for our paper.

| symbol | meaning | symbol | meaning |
|---|---|---|---|
| $\boldsymbol{\alpha}_i$ | $i$th channel embedding of inputs MTS | $\boldsymbol{\mu}_i, \boldsymbol{\sigma}_i$ | statistics of $\boldsymbol{\alpha}_i$ |
| $s(u,v)$ | symmetric similarity function of $u$ and $v$ | $\boldsymbol{Q}_i, \boldsymbol{K}_i, \boldsymbol{V}_i$ | query, key and value of the $i$th head |
| $\boldsymbol{H}_i$ | output of the $i$th head | $\tilde{\boldsymbol{A}}, \boldsymbol{A}$ | (normalized) symmetric adjacent matrix |
| $\tilde{\boldsymbol{h}}, \boldsymbol{h}$ | (dynamic) latent states | $\tilde{\boldsymbol{z}}_{t,n}, \boldsymbol{z}_{t,n}$ | (Convolutional) latent variable at each timestep |
| $\mu_{t,n}, \boldsymbol{\sigma}_{t,n}$ | statistics of $\boldsymbol{z}_{t,n}$ | $\boldsymbol{x}_{t,n}$ | generated MTS at each timestep |
| $\mu_{t,n}^x, \boldsymbol{\sigma}_{t,n}^x$ | statistics of $\boldsymbol{x}_{t,n}$ | $\boldsymbol{\beta}$ | mapping matrix between $\boldsymbol{h}_{t,n}$ and $\boldsymbol{x}_{t,n}$ |
| $\tilde{\boldsymbol{h}}_n^z, \tilde{\boldsymbol{h}}_n^x$ | dynamic latent states of $\boldsymbol{z}$ and $\boldsymbol{x}$ | $\tilde{\boldsymbol{h}}_n$ | concatenated dynamic latent states |
| $\boldsymbol{O}_n^x, \boldsymbol{O}_n^z$ | output of MSA block for $\boldsymbol{z}$ and $\boldsymbol{x}$ | $\boldsymbol{O}$ | output of MSA block |

types of services, e.g., Video on Demand (VoD) or live streaming video, etc. Besides, for each website, KPIs span about one and a half months and are collected every 60 seconds. In our experiments, for each website, the first half of the KPIs are used for training, while the second half are used for testing. Note that the ground-truth anomalies in the test set of CDN have been confirmed by human operators. For the public datasets, Server Machine Dataset (SMD) is a real-world public dataset Su et al. (2019) that contains data from 28 server machines that are monitored by 38 KPIs individually. In addition, for each server machine, KPIs span about five weeks. Mars Science Laboratory (MSL) rover Dataset is also a real-world public and expert-labeled dataset from NASA Hundman et al. (2018) containing the data of 27 entities each monitored by 55 metrics. Note that the other Soil Moisture Active Passive (SMAP) satellite dataset is also released by NASA Hundman et al. (2018), and both MSL and SMAP are collected from spacecraft where the first dimension is the value of telemetry channel, while the rest dimensions are command information that encoded as 0 or 1.

**Forecasting:** For forecasting task, we use four datasets, including {ETTh1, ETTh2} and {ETTm1, ETTm1}, created in Zhou et al. (2021) based on Electricity Transformer Temperature (ETT), which is a crucial indicator in the electric power long-term deployment. ETT used here is created by collecting 2 years data from two separated counties in China. To explorer the granularity on the LSTF problem, Zhou et al. (2021) creates separate datasets as {ETTh1, ETTh2} for 1-hour level and {ETTm1, ETTm2} for 15-minutes-level. Each data point consists of the target value "oil temperature" and 6 power load features. The train/val/test is 12/4/4 months, which is the same setting as in Zhou et al. (2021). Weather is the dataset that contains local climatological data for nearly 1,600 U.S. locations, 4 years from 2010 to 2013, where data points are collected every 1 hour. Each data point consists of the target value "wet bulb" and 11 climate features. The train/val/test is 28/10/10 months. ECL (Electricity Consuming Load) dataset collects the electricity consumption (Kwh) of 321 clients. Due to the missing data, we use the data released by Zhou et al. (2021) that converts the dataset into hourly consumption of 2 years and set 'MT 320' as the target value. The train/val/test is 15/3/4 months.

Table 5: Basic statistics of anomaly detection datasets

| Statistics | Anomaly Detection | | | |
|---|---|---|---|---|
| | **CDN** | **SMD** | **MSL** | **SMAP** |
| Dimensions | 12*36 | 28*38 | 27*55 | 55*25 |
| Granularity (sec) | 60 | 60 | 60 | 60 |
| Training set size | 344,843 | 708,405 | 58,317 | 135,181 |
| Testing set size | 344,844 | 708,420 | 73,729 | 427,617 |
| Anomaly ratio (%) | 3.44 | 4.16 | 10.72 | 13.13 |

Table 6: Basic statistics of forecasting datasets.

| Statistics | Forecasting | | | | | | | | | | | |
|---|---|---|---|---|---|---|---|---|---|---|---|---|
| | **ETT**$^*$ | | | | **ECL** | | | | **Weather** | | | |
| | 96 | 168 | 288 | 336 | 96 | 168 | 288 | 336 | 24 | 48 | 96 | 168 |
| **Channels** | 7 | | | | 321 | | | | 21 | | | |
| **Training set size** | 8449 | 8377 | 8257 | 8209 | 18221 | 18149 | 18029 | 17981 | 36768 | 36744 | 36696 | 36624 |
| **Validation set size** | 2785 | 2713 | 2593 | 2545 | 2537 | 2465 | 2345 | 2297 | 5247 | 5223 | 5175 | 5103 |
| **Testing set size** | 2785 | 2713 | 2593 | 2545 | 5165 | 5093 | 4973 | 4925 | 10516 | 10492 | 10444 | 10372 |

$*$ ETT$^*$ means the full benchmark of ETTh1, ETTh2, ETTm1 and ETTm2.

## A.5 EXPERIMENT SETTING

**Hyper-parameters:** The hyper-parameters of data preprocessing are set as $T = 20$, $o = 4$ for anomaly detection and $T = 20$, $o = 0$ for forecasting. We employ Adam optimizer with learning rate of $0.0002$, batch size of $64$, total training epochs of $90$ for all tasks. The balance factor $\gamma$ for anomaly detection and forecasting are set as $0.1$ and $0.5$, respectively. The length of forecasting is set to be $12$, $24$, $48$, $96$, and $168$ in turn.

**Model architecture:** For anomaly detection experiment, the dimensions of channel embeddings for inputs $\boldsymbol{\alpha}_i$ are all set as $256$, that is $d = 256$. The dimension of mapping matrix $\boldsymbol{\beta}$ is set as $256 \times 10$, that is $K^{'} = 10$. We employ a deconvolutional neural network (DCNN) and CNN in generation and inference process separately, both CNN encoder and DCNN decoder are with 3 convolutional layers, whose filters and strides are set according to the number of timesteps of the observed variables. Then, the latent variables and inputs are feed into MSA blocks, respectively. Thirdly, the aggregated dynamic information $\tilde{\boldsymbol{h}}_n$ of inputs $\boldsymbol{x}$ and latent variables $\boldsymbol{z}$ can be calculated by AGCN blocks. And the architectures of MSA and AGCN are shown in Table. 7, where $V$ represents the inputs' dimension, it various from different datasets as discussed in A.6. Finally, we employ a well-developed Transformer architecture Li et al. (2019) to derive the dynamic latent states $\boldsymbol{h}$. The specific architecture can be found in Table. 8. The head of VG-Trans is set as $8$. For forecasting experiment, we select the hyperparameters, including embedding size, latent size, latent dimension and so on, with validation data.

Table 7: MSA and AGCN blocks for inputs and latent variables

| Layer | Size | Norm/Act. |
|---|---|---|
| Input | 25/$V$ | |
| Linear_Query | 25/$V$ | |
| Linear_Key | 25/$V$ | |
| Identity_Value | 25/$V$ | |
| Linear+Add | 25/$V$ | BN |
| GCN | using $\boldsymbol{\alpha}$ | |
| Linear+Add | 25/$V$ | BN |

Table 8: Transformer for dynamic latent states

| Layer | Size | Norm/Act. |
|---|---|---|
| Input | 25+V | |
| PosEmbd | 256 | |
| Linear_Query | 25+V+256 | |
| Linear_Key | 25+V+256 | |
| Linear_Value | 25+V+256 | |
| Linear+Add | 25+V+256 | BN |
| GCN | using $\alpha$ | |
| Linear+Add | 25+v+256 | BN |

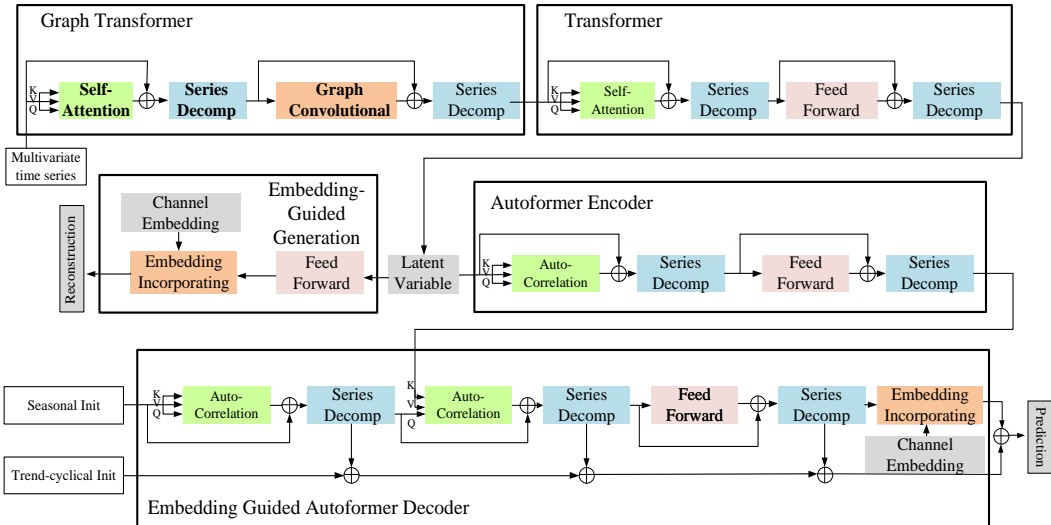

Figure 8: The detailed structure of VG-Autoformer.

## A.6 MODEL PROPERTIES

Comparing with previous Transformer based methods for modeling MTS, our proposed VG-Trans has the following properties:

**1):** Introducing Gaussian-distributed channel embeddings, modeling the stochasticity of different sensors and their dependencies in MTS;

**2):** Introducing graph network into transformer structure for capturing the diverse relationships between different channels within MTS;

**3):** Enhancing generalization capacity by incorporating channel embeddings into generative process;

**4):** Being optimized with both prediction and reconstruction losses, VG-Trans can explore both the periodic information and global distribution of MTS for extracting expressive representations;

**5):** The prior of the latent variable in EPM is conditioned on the previous latent states of G-Trans for captureing temporal dependency, thus to help learning richer representations.

All of these properties ensure more expressive and robust representations of MTS and enhance generative capacity for complex MTS, thus achieves accurate detection and forecasting.

## A.7 VARIATIONAL GRAPH AUTOFORMER

We claim that the proposed Variational Graph Transformer (VG-Trans) is a flexible framework having better compatibility with other methods, such as Anomaly-Trans (please refer to results of Table 1 in the text) and Autoformer Wu et al. (2021). In this section, we mainly discuss how to extend the VG-Trans with the help of Autoformer, the pipeline of which is illustrated in Fig. 8.

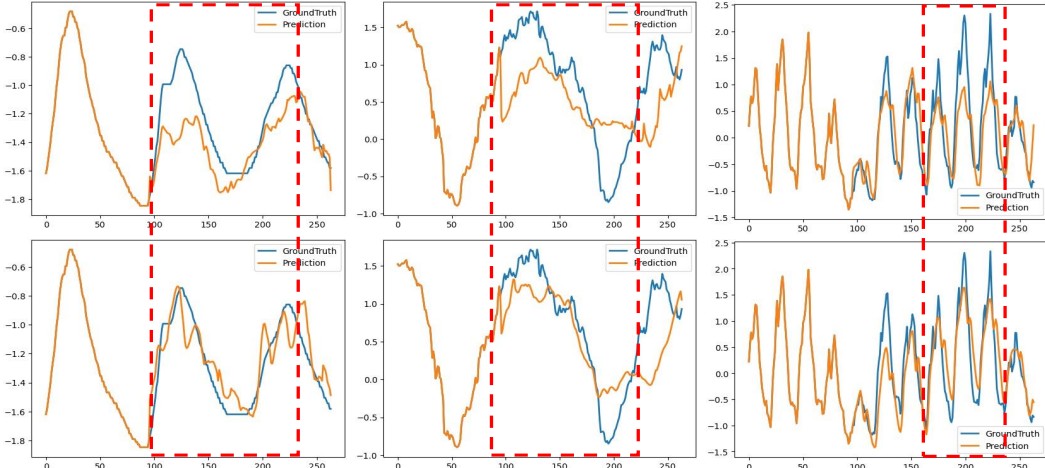

Figure 9: Prediction cases of simple MTS from the ETT, Weather and ECL datasets (from left to right). **Top:** results of Autoformer; **Bottom:** results of VG-Autoformer.

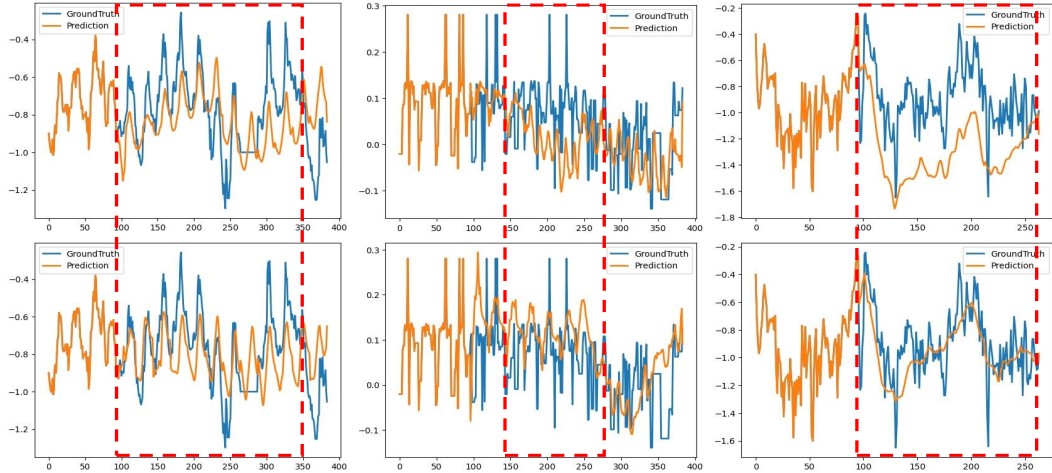

Figure 10: Prediction cases of complex MTS from the ETT dataset. **Top:** results of Autoformer; **Bottom:** results of VG-Autoformer.

Specifically, we replace the standard Transformer with Autoformer encoder. Then we add Autoformer decoder for forecasting task while keep the VG-Trans decoder and Graph Transformer unchanged. We find that it is important to keep the `Seasonal Init` and `Trend-cyclical Init` directly derived from the raw input $x_n$ to maintain Autoformer's capability for forecasting. On the other hand, considering full ELBO and channel-wise relationships enhances the robustness of capturing long-term dependencies, thus achieves better prediction results, as shown in Table. 2, in which u/v represents the author reported result u and our reproduced result v.

## A.8 MORE VISUALIZATION RESULTS

We illustrate more prediction cases of simple and complex MTS on forecasting task in Fig. 9 and Fig. 10, respectively. And the improvement areas are highlighted with red dotted boxes. Our model gives the best performance on different datasets compared with Autoformer. Specifically, VG-Autoformer complements more missing details in Autoformer thanks to its design of capturing both temporal and channel dependencies at the same time.

## A.9 BALANCE BETWEEN PREDICTION AND RECONSTRUCTION

As discussed in Sec. 3, we combine prediction and reconstruction losses on VG-Trans, which enables our models to learn expressive representations efficiently, and introduce a parameter $\gamma$ to balance

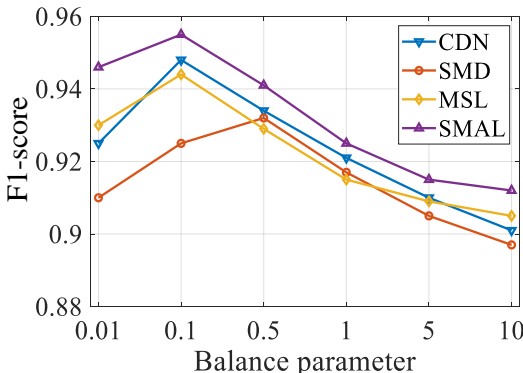

Figure 11: The influence of balance parameter $\gamma$ on anomaly detection task.

the effect of them. Here, we evaluate the influence of $\gamma$ to anomaly detection task, the results are reported in Fig. 11. As we can see, excessively small and large $\gamma$ will lead to weaker performance, illustrating the effectiveness of both reconstruction and prediction losses. Besides, relatively small weight for prediction loss is good for detection for exploring the global distribution of the MTS.

