# OpenReview forum: "VARIATIONAL ADAPTIVE GRAPH TRANSFORMER FOR MULTIVARIATE TIME SERIES MODELING"
_ICLR.cc/2023/Conference — Submitted to ICLR 2023_

### Official Review · Reviewer_XSSh · 2022-10-22

**Confidence:** 2
**Correctness:** 3
**Technical Novelty And Significance:** 3
**Empirical Novelty And Significance:** 3
**Recommendation:** 6

**Clarity, Quality, Novelty And Reproducibility:**

The paper is clearly written and easy to follow.
The proposed idea looks novel to me.
For reproducibility, the authors provide code but there's no documentation on how to run the code.

**Strength And Weaknesses:**

Strengths:
- The proposed idea looks novel to me, though I'm not very familiar with related work in this subfield.
- The authors conducted thorough experiments and the results look competitive.

Weaknesses:
- It's better to include std in the experiments to help understand how significant the improvements are.
- In Table 3, it is a bit confusing to me why the reconstruction loss leads to better results than prediction loss. Why is it the case?
- Why there are some entries unavailable in Table 3?

**Summary Of The Paper:**

The authors propose a G-Trans module, which incorporates channel-relationship learning into the Transformer structure for multivariate time series modeling. The authors further added  a VAE-structured module to model the non-deterministic within both temporal
and cross-channel dependencies of MTS. Experiments on both anomaly detection and forecasting tasks illustrate the efficiency of our
model on MTS modeling.

**Summary Of The Review:**

A novel approach on MTS by introducing graph structure and VAE encoder.

---

### Official Review · Reviewer_Z2ri · 2022-10-24

**Confidence:** 4
**Correctness:** 3
**Technical Novelty And Significance:** 2
**Empirical Novelty And Significance:** 2
**Recommendation:** 5

**Clarity, Quality, Novelty And Reproducibility:**

The paper is mostly well-organized. The techniques are solid and correct. The code is provided for reproducibility.

**Strength And Weaknesses:**

Pros:
1. The problem studied in this paper is interesting and practical.
2. The paper is well-structured.
3. The experiments verify the effectiveness of the method.

Cons:
1. I suggest that the author should further clarify the first contribution (i.e., the G-Trans module). Since incorporating relationship learning between different time series have actually been explored by many works in the field of spatial-temporal forecasting. What’s the difference in the proposed G-Trans ?
2. The term “dynamic” is mentioned many times in the paper, with no clear explanation. For example, in related works, “However, these methods are still all non-dynamic or RNN based models, limiting their power in capturing complex relationships across long-distance time steps”
3. The authors claim that “VG-Trans is able to get the robust representation of MTS”. How to prove it?
4. In Figure 2(c), the authors use Graph Convolution network to replace the first Feed Forward Network, by stating that GCN can better capture the diverse relationship than FFN. If so, why not use GCN to replace all FFN in Transformer? Will the result get better by doing so?


**Summary Of The Paper:**

The authors propose an approach to model the time dependencies, cross-channel dependencies, and stochasticity for multivariate time series (MTS) modeling. Specifically, an adaptive graph Transformer is first designed to learn the cross-channel relationships in MTS. To model the stochasticity, a channel embedding guided probabilistic generative module is further proposed. Lastly, the authors propose an autoencoding inference scheme that combines prediction and reconstruction loss for better representation learning. Experiments on the time series forecasting and anomaly detection task demonstrate the effectiveness of the proposed method.

**Summary Of The Review:**

Given the strength and weaknesses mentioned above, this paper is below the level of ICLR and needs to be improved.

---

### Official Review · Reviewer_RixC · 2022-10-27

**Confidence:** 3
**Correctness:** 2
**Technical Novelty And Significance:** 2
**Empirical Novelty And Significance:** 2
**Recommendation:** 3

**Clarity, Quality, Novelty And Reproducibility:**

The organization of this manuscript is clear.

Some claims and motivations lack theoretical support or explanation. Thus the overall quality of the paper is not high.

The novelty of the idea in this work is very limited, and the proposed method is not very insightful.

**Strength And Weaknesses:**

Strength: The organization of this manuscript is clear and it's easy to read. The proposed approach performs better in anomaly detection and time series forecasting tasks compared to some baselines.

Weaknesses:

--Some claims and motivations lack theoretical support or explanation. E.g., it is hard to understand that "However, Transformers in general do not well capture the diverse relationships between different channels within MTS and have difficulty in modeling MTS with complex distributions due to the lack of stochasticity." and "As probabilistic methods, VG-Trans realizes smoother anomaly scores than previous methods for considering stochasticity within MTS."

--The novelty of the idea in this work is very limited, and the proposed method is not very insightful. It is a natural idea to use graph structure in time series, and many works have tried, such as
[1] Graph neural network-based anomaly detection in multivariate time series, in: Proceedings of the AAAI Conference on Artificial Intelligence, 2021.
[2] MST-GAT: A multimodal spatial–temporal graph attention network for time series anomaly detection. Information Fusion 2023.
Just changing RNN to Transformer is not inspiring.

--The motivation for using probabilistic modeling and variational inference is also unclear, and no experimental results have shown the existence of stochasticity and the advantage of elimination it.

--The experimental comparison is still superficial, and only focuses on whether the performance is improved, without in-depth analysis and experimental verification of the claimed assumptions about the advantage of modeling channel relationship and modeling stochasticity.

**Summary Of The Paper:**

This paper proposes to combine graph structure with transformer to model multivariate time series, and use probabilistic modeling method and variational inference to remove random noise in time series. The experiments on the task of anomaly detection and time series forecasting show the effectiveness of the proposed method.

**Summary Of The Review:**

This work is just the fusion and utilization of existing techniques to realize multivariate time series modeling. The innovation of the method is limited, the experimental design is not fine enough, and no enlightening insights are given.

---

### Decision · Program_Chairs · 2023-01-20

**Decision:**

Reject

**Justification For Why Not Higher Score:**

Reviewers have concerns on the contributions

**Justification For Why Not Lower Score:**

N/A

**Metareview: Summary, Strengths And Weaknesses:**

This paper has proposed an approach to model the time dependencies, cross-channel dependencies, and stochasticity for multivariate time series (MTS) modeling.  It performed experiments on tasks including time series forecasting and anomaly detection which shows its  effectiveness.

Some contributions are not well justified and some claims are not proved.

**Summary Of Ac-Reviewer Meeting:**

No need as reviewers have provided lower rating for this paper consistently